# Ingested Polystyrene Nanospheres Translocate to Placenta and Fetal Tissues in Pregnant Rats: Potential Health Implications

**DOI:** 10.3390/nano13040720

**Published:** 2023-02-14

**Authors:** Chelsea M. Cary, Glen M. DeLoid, Zhenning Yang, Dimitrios Bitounis, Marianne Polunas, Michael J. Goedken, Brian Buckley, Byron Cheatham, Phoebe A. Stapleton, Philip Demokritou

**Affiliations:** 1Nanoscience and Advanced Materials Center, Environmental and Occupational Health Sciences Institute (EOHSI), School of Public Health, Rutgers University, Piscataway, NJ 08854, USA; 2Ernest Mario School of Pharmacy, Rutgers University, Piscataway, NJ 08854, USA; 3Center for Nanotechnology and Nanotoxicology, Department of Environmental Health, Harvard T. H. Chan School of Public Health, Boston, MA 02115, USA; 4Research Pathology Services, Rutgers University, Piscataway, NJ 08854, USA; 5CytoViva, Inc., Auburn, AL 36832, USA

**Keywords:** plastics, nanoplastics, rat, translocation, placenta, fetus

## Abstract

Recent studies in experimental animals found that oral exposure to micro- and nano-plastics (MNPs) during pregnancy had multiple adverse effects on outcomes and progeny, although no study has yet identified the translocation of ingested MNPs to the placenta or fetal tissues, which might account for those effects. We therefore assessed the placental and fetal translocation of ingested nanoscale polystyrene MNPs in pregnant rats. Sprague Dawley rats (N = 5) were gavaged on gestational day 19 with 10 mL/kg of 250 µg/mL 25 nm carboxylated polystyrene spheres (PS25C) and sacrificed after 24 h. Hyperspectral imaging of harvested placental and fetal tissues identified abundant PS25C within the placenta and in all fetal tissues examined, including liver, kidney, heart, lung and brain, where they appeared in 10–25 µm clusters. These findings demonstrate that ingested nanoscale polystyrene MNPs can breach the intestinal barrier and subsequently the maternal–fetal barrier of the placenta to access the fetal circulation and all fetal tissues. Further studies are needed to assess the mechanisms of MNP translocation across the intestinal and placental barriers, the effects of MNP polymer, size and other physicochemical properties on translocation, as well as the potential adverse effects of MNP translocation on the developing fetus.

## 1. Introduction

After three quarters of a century of increasing production of and reliance on plastic materials, six billion metric tons of plastic waste have been deposited in the environment [1]—enough to coat the entire surface of the Earth in a plastic film over 10 µm thick, the thickness of a plastic grocery produce bag. Through various natural, commercial and municipal degradation processes, all plastics are eventually fragmented into micrometer- and then nanometer-scale plastic particles and fibers known as micro- and nanoplastics (MNPs), as discussed in more detail elsewhere [1,2,3,4,5,6,7]. Consequently, MNPs have now become a nearly ubiquitous contaminant of both the environment and the food web [1,5,6,8,9,10,11,12], to the extent that the average human now ingests an estimated 5 g of MNPs per week [12].

Results from previous in vitro and in vivo studies suggest that ingested MNPs can cross the intestinal barrier to access the circulatory system, and thence the peripheral tissues. Walczak et al. reported the size- and surface-chemistry-dependent uptake of MNPs in multiple in vitro intestinal cell models [13]. Likewise, we found that 25 nm, but not 100 or 1000 nm, carboxylated polystyrene (PS) MNPs were readily taken up by and translocated across an in vitro triculture small-intestinal epithelium [2]. Several studies in laboratory animals have also demonstrated the intestinal uptake of ingested PS MNPs, with translocation to multiple tissues and organs [14,15,16,17,18], including Peyer’s patches, lymph nodes, liver, spleen and blood in rats [15,16,17]; liver, kidney and gut in mice [18] and brain in fish [14]. In addition, MNPs have recently been identified in human tissues, including human colectomy specimens [19], blood from human volunteers which contained multiple highly produced polymers with a mean total blood concentration of 1.6 µg/mL [20], and both maternal and fetal zones of human placentas [21]. The latter findings suggest the potential for maternal–fetal transfer of MNPs, and are consistent with our recent report of the translocation and deposition of 20 nm PS spheres in fetal tissues 24 h after intratracheal instillation in pregnant rats [22]. However, the exposure routes responsible for the MNPs found in human tissues are not known, and given the aforementioned findings of intestinal MNP uptake and the current estimated human MNP ingestion rate, are most likely a combination of both ingestion and inhalation. Moreover, recent studies in experimental animals found that oral exposure to MNPs during pregnancy had multiple adverse effects on pregnancy outcomes and progeny [23,24,25,26,27,28], suggesting potential fetal translocation of the ingested MNPs. To date, however, no study has verified the translocation of ingested MNPs to the placenta or fetal tissues, which might account for those effects.

We therefore hypothesized that ingested nanoscale PS MNPs could translocate across the intestinal epithelium to reach the circulation, and subsequently cross the maternal–fetal barrier of the placenta to reach the fetal circulation and tissues. In the present study, we tested this hypothesis in a pregnant rat model, as detailed below.

## 2. Materials and Methods

### 2.1. Study Design Overview

To assess the placental and fetal translocation of MNPs, we gavaged pregnant (gestational day 19—GD19) female Sprague Dawley rats with 10 g/kg of a 250 µg/mL suspension of 25 nm carboxyl-modified PS spheres (PS25C) and harvested placentas and fetal tissues after 24 h (GD20) for analysis using enhanced darkfield hyperspectral microscopy (EDHM), which allows for the identification and localization of materials within tissues based on their specific hyperspectral signal. An overview of the study design is shown in Figure 1.

### 2.2. Primary MNP Materials

Primary carboxyl-modified red fluorescent PS nano-microspheres with sizes of 25 nm (PS25C) were obtained from Thermo, Inc. (Waltham, MA, USA).

### 2.3. Preparation and Characterization of the Dispersion of Primary Polystyrene MNPs

Carboxyl-modified red fluorescent FluoSpheres™ were obtained from Thermo, Inc. (Waltham MA, USA). The nominal size of the spheres was 20 nm (25 nm according to the manufacturer’s characterization for the product lot used). MNPs from Thermo were diluted in sterile deionized water to the desired starting MNP concentration and vortexed for 20 s.

Water suspensions of PS25C used for gavage were analyzed by dynamic light scattering (DLS) for the determination of hydrodynamic diameter (*d*_H_) and polydispersity index (PdI), and by electrophoretic light scattering (ELS) to determine zeta potential (ζ) and conductivity (σ) using a Zetasizer Nano-ZS (Malvern Instruments, Ltd.) as previously described [29].

### 2.4. Selection of MNP Concentration

The oral gavage MNP concentration used in this study was based on the estimated weekly human intake of 5 g, primarily from drinking water [30]. Assuming a mean adult total water intake of ~3.1 L/d, based on an average across age groups in data from a recent study of water and beverage consumption in the U.S. [31], 5 g per week would correspond to a mean ingested MNP concentration of ~230 µg/mL. We therefore chose a concentration of 250 µg/mL for this study. Given the expected continued increase in plastic production and waste generation, and the fact that technical limitations have prevented the inclusion of nanoscale MNPs in most environmental and exposure studies to date, we believe this is a reasonable starting dose for the assessment of the placental and fetal transfer of ingested MNPs in water.

### 2.5. Exposure and Sample Collection

Time-pregnant Sprague Dawley rats purchased from Charles River Laboratories were received on gestational day (GD) 17–18 and randomly divided into control and exposure groups. Exposure group animals were lightly anesthetized with isoflurane gas (5% induction) and gavaged with a bolus dose of MNPs (250 µg/mL, 10 mL/kg) using a 20-gauge 4-inch stainless-steel ball-tipped animal feeding needle on GD 19. Control animals remained untreated. Rats were monitored after gavage until they had regained consciousness and normal physiological activity (e.g., walking, eating, drinking, grooming, resting). Exposed and control animals were sacrificed 24 h following exposure per approved Rutgers’ IACUC protocols. Maternal tissues, periparturient fetuses and fetal tissues (liver, kidney, lung, heart and brain) were harvested, fixed in 10% formalin and embedded in paraffin. Maternal weight, litter number, fetal weight, placental weight, placental efficiency and resorption number were recorded.

### 2.6. Enhanced Darkfield Hyperspectral Microscopy (EDHM) of Placenta and Fetal Tissues

To prepare specimens for EDHM analysis, thin sections (5 µm) of formalin-fixed, paraffin-embedded placental and fetal tissues were deparaffinized by washing three times for 5 min in xylene, then rehydrated through a series of washes in 100, 95, 70 and 50% ethanol (2–10 min washes each), followed by 2–5 min washes in deionized water and finally washed three times in PBS. Slides were then air dried and mounted with SlowFade Diamond mountant (Thermo, Inc.) and a #1.5 coverslip.

A CytoViva Enhanced Darkfield Hyperspectral Microscope (CytoViva, Inc., Auburn, AL, USA) was used to identify and visualize PS25C within placental and fetal tissue sections. Enhanced darkfield hyperspectral microscopy (EDHM) allows imaging of nanoscale sample elements based on their scattering or emission properties. The system utilizes a tungsten halogen light (Dolan Jenner, Boxborough, MA, USA) illumination source with a variable power (0–150 W) control, which was set to the full 150 W output for these experiments. Light from this light source is directed to the microscope via a liquid light guide (Newport Inc., Newport, CA, USA). The light guide is connected to the enhanced darkfield illuminator system, which is mounted onto the condenser mount of the research-grade optical microscope. The illuminator system consists of an annular cardioid oil condenser, which produces highly collimated light at oblique angles with a numerical aperture of 1.2–1.4 and collimating lenses, which together modify the geometry of the source light to closely match that of the system’s cardioid oil condenser. The collimating lenses focus the light onto the condenser to refine and fix the Koehler illumination of the light onto the condenser, allowing the highly oblique darkfield illumination to be focused onto the precise focal plane of the sample without losing the integrity of the Koehler illumination. This indirect oblique illumination allows collection of the reflected or elastically scattered light from the sample, which permits the visual differentiation of objects with refractive indices similar to that of the background.

In the EDHM system, the scattered light image from the objective is projected onto a visible and near-infrared (VNIR) diffraction grating spectrograph (Specim, Oulu, Finland), which separates the light into distinct wavelengths from 400 to 1000 nm with a spectral resolution of 2 nm. This spectrally resolved light is then projected through a 30 µm slit onto the pixels of a charge-coupled device (CCD) video camera (PCO, Kelheim, Germany). This camera then sends the data to a highly custom version of ENVI 4.8 hyperspectral image capture analysis software. The hyperspectral image is captured in a pixel row by pixel row line scan using an automated translational stage (Prior Scientific Instruments Ltd., Cambridge, UK) with a 10 nm step resolution. The hyperspectral image is then built, pixel row by pixel row.

Analysis of resulting hyperspectral images was conducted using a customized version of ENVI 4.8 hyperspectral image analysis software (Harris Geospatial Solutions, Inc., Herndon, VA, USA). Each pixel of the hyperspectral image contains the optical spectral response from 400 to 1000 nm of that pixel’s spatial area at a 2 nm spectral resolution. Image analysis was conducted to spectrally map labeled polystyrene particles in the tissue. Scattered light was collected using a 60x oil iris objective (Olympus, Inc.). Thousands of individual pixel spectra were captured to create a spectral library from tissues from pups of dams exposed to PS25C nanoparticles. These spectral libraries were then compared to all pixels in the negative control tissue (from pups of control dams) image in a process known as “Filter Spectral Library”. A spectral mapping algorithm known as “Spectral Angle Mapper” (SAM) was then utilized to match pixels to the reference spectrum to spectrally map the polystyrene in tissue.

## 3. Results

### 3.1. Characterization of MNP Suspension in Water

The water suspension of PS25C was analyzed by dynamic light scattering (DLS) and electrophoretic light scattering (ELS), which revealed a monomodal particle dispersion with an average hydrodynamic diameter of 33.02 ± 0.01 nm, polydispersity index of 0.151 ± 0.013, zeta (ζ)-potential of −66.8 ± 1.76 mV and conductivity (σ) of 0.0414 ± 0.0004 mS/cm. Intensity- and volume-weighted size distributions are shown in Figure 2.

### 3.2. Translocation of 25 nm Carboxylated Polystyrene MNPs to Placenta Following Ingestion Exposure in Pregnant GD19 Sprague Dawley Rats

Hyperspectral images of 4 µm sections of placentas from PS25C-exposed dams are shown in Figure 3. Hyperspectral signals identified as PS25C were observed in large irregular clusters with maximum diameters of up to ~100 µm, as well as in smaller discrete locations within the placental tissue and blood. Although definitive identification of the placental zones in Figure 3a–c is not possible based on the darkfield images alone, the general morphology in these panels is consistent with decidua, junctional zone and labyrinth zone, respectively. Multiple PS25C signals can be seen associated with red blood cells (RBCs), whose ellipsoid outlines can be seen within the blood sinuses in the presumed labyrinth zone in Figure 3c. From the appearance of a strip of epithelium containing a vessel filled with RBCs, the region in Figure 3d can be identified as chorionic plate and the large vessel as fetal. Several PS25C signals can be seen associated with RBCs within the fetal vessel and within the vessel endothelium.

### 3.3. Translocation of 25 nm Carboxylated Polystyrene MNPs to Fetal Tissues following Maternal Ingestion Exposure

To assess the potential fetal tissue translocation of ingested PS25C, EDHM was used to identify PS25C MNP signals within sections of fetal organs harvested 24 h after oral gavage. Large clusters of PS25C were identified in all gestationally exposed fetal tissues examined, including liver, kidney, lung, heart and brain (Figure 4). The fetal tissue clusters of PS25C MNPs were roughly ovate, ranging from about 10 to 25 µm in size, with irregular edges.

### 3.4. Effect of Maternal Ingestion Exposure to 25 nm Carboxylate-Modified Polystyrene (PS25C) on Maternal, Placental and Litter Characteristics

The placenta is a well-known barrier, but also functions as a vascular and endocrine organ to enable fetal development. Because such functions could potentially be impaired by the presence of MNPs, we assessed changes in critical maternal and litter characteristics, including maternal weight, litter size, fetal weight, placental weight, placental efficiency and total number of sites of resorption. The results of these measurements in control and treated rats are reported in Table 1. Placenta weights were significantly lower (by ~18%) in the exposed group compared with control, while there was no significant difference in fetal weights. Placental efficiency, defined as grams of fetus per gram of placenta, was thus increased by ~22% in the exposed group.

### 3.5. Histopathological Analysis of Fetal Tissues

Histopathological examination of fetal liver, kidney, lung, brain and heart tissues from three pups from each control and PS25C-treated dam revealed no notable findings in any of the tissues and no notable differences between tissues from pups of treated and untreated dams.

## 4. Discussion

The results of this in vivo study confirm our hypothesis that ingested nanoscale MNPs can (1) translocate across the intestinal epithelium to reach the circulation and (2) cross the maternal–fetal barrier of the placenta to reach fetal circulation and tissues. These findings strongly suggest the potential for the maternal–fetal transfer of nanoscale PS MNPs.

In this experimental model, the PS MNPs must cross two physiological barriers to reach the fetal compartment: the gastrointestinal tract and the placenta. As described in the Introduction, studies by our lab and others have found that PS MNPs can cross the intestinal epithelium and distribute to multiple organs. Further studies are needed to determine the specific mechanisms involved in MNP translocation across the intestinal epithelium, which could include passive transcellular and/or paracellular diffusion as well as one or more active (energy-dependent) endocytosis mechanisms, recently reviewed by Rennick et al. [32]. Such studies are underway in our lab, and will be reported in a future companion manuscript. Moreover, pregnancy hormones (i.e., progesterone), which would be elevated in the pregnant rat model used in this study, are known to cause smooth muscle relaxation, slowing digestion and prolonging gastrointestinal transit, thereby increasing the likelihood of MNP uptake. Further studies are therefore also needed to assess the role of progesterone and other pregnancy hormones on MNP uptake by the intestine.

The second physiological barrier required for translocation to fetal tissues is the maternal–fetal barrier, which occurs in the placental labyrinth zone. In rats this consists of four distinct layers: a layer of cytotrophoblasts, a double layer of syncytiotrophoblasts and the fetal capillary endothelium (the arrangement is somewhat different in humans, as discussed below). Because the cytotrophoblast layer is discontinuous and the fetal capillary endothelium is fenestrated, they do not contribute substantially to barrier function. The two continuous, uninterrupted syncytiotrophoblast layers, which are connected at gap junctions through Connexin 26, act functionally as a single syncytial layer to provide barrier function [33]. The appearance of PS25C signals within a fetal vessel in the chorionic plate (Figure 3d) and within the fetal tissues (Figure 4) suggests that this cellular barrier can be breached by these MNPs. We recently reported a similar translocation and deposition of 20 nm PS spheres in multiple maternal organs and fetal tissues within 24 h following pulmonary intratracheal instillation in pregnant rats [22]. It is worth noting that while in rats the maternal–fetal interface is hemotrichorial (i.e., composed of a layer of cytotrophoblasts and two fused layers of syncytiotrophoblasts), in humans the maternal–fetal interface is hemomonochorial, consisting of a single layer of mixed cytotrophoblasts and syncytiotrophoblasts [33]. Given this anatomical difference, the human placental maternal–fetal barrier could be more permissive, allowing greater passage of MNPs to the fetus. Moreover, MNPs were recently identified in both the maternal and fetal zones of human placental samples after real-world exposure [21].

The presence 24 h post gavage of PS25C within all fetal tissues examined from pups of exposed dams, including liver, kidney, lung, heart and brain (Figure 4), is consistent with the appearance of PS25C within the fetal vessel in the chorionic plate (Figure 3d), which demonstrated translocation of the ingested PS25C to the fetal circulation. This further suggests that MNPs reaching the fetal circulation can cross fetal blood–tissue borders and extravasate into the tissues of all of these organs. The occurrence of PS25C MNPs within 10–25 µm roughly elliptical clusters with irregular borders suggests the possibility that PS25C MNPs may have been taken up by fetal tissue macrophages. Further studies, such as immunofluorescence staining and microscopy to evaluate colocalization of the hyperspectral PS25C signals and macrophage or other cell markers, are needed to determine what cells (if any) these clusters may represent. However, if the observed clusters are in fact fetal tissue macrophages filled with MNPs, this would raise concerns about the impact on the health and function of those macrophages, which play critical roles in tissue remodeling and repair, angiogenesis, innate immunity and inflammation during development [34]. Altered fetal tissue macrophage function has been implicated in a number of neonatal and perinatal disorders, including hypoxic-ischemic encephalopathy (HIE), bronchopulmonary dysplasia (BPD) and necrotizing enterocolitis (NEC) [34]. In addition, several recently published studies have reported a variety of adverse effects on the fetus and progeny after oral exposure to polystyrene MNPs in pregnant mice, including metabolic disorders [27,28], hepatic and testicular toxicity [25], fetal growth restriction [24] and multiple developmental brain abnormalities accompanied by neurophysiological and cognitive deficits [23]. Further studies are needed to quantify the MNP concentrations and doses in the placenta and fetal organs as a function of time in a variety of exposure scenarios in order to assess the potential effects of the presence of MNPs in fetal tissues on fetal health and development, and to potentially mechanistically link the adverse effects observed in the offspring of rats exposed during pregnancy to the presence and specific locations of MNPs in fetal tissues.

A significant decrease in placental weight was also observed in PS25C-treated dams compared to untreated controls, while no significant difference was seen in fetal weights between the groups, resulting in an increase in calculated placental efficiency in the treated dams (Table 1). However, because control animals in this pilot study did not undergo gavage under anesthesia it is possible that the observed differences in placental weights were caused by physiological stress responses to the anesthesia or gavage procedure. Further studies with a sham gavage control are therefore needed to validate and further evaluate the possible significance of these findings.

Finally, it is worth noting that simplistic models of MNPs such as the PS used in this study are not environmentally relevant. More environmentally relevant MNPs which are by-products of the thermal, mechanical and photo-induced degradation of polymers in the environment are needed for future toxicological assessment studies. It is well known that the physicochemical properties of nanoscale materials define their cellular uptake and biodistribution [35,36]. Such environmentally relevant MNPs may have completely different biokinetics and bioactivity outcomes. For example, the weathering (photo-aging) of plastics in the environment results in the formation of surface carboxyl and carbonyl functional groups [4], which could dramatically alter the biointeractions of otherwise hydrophobic MNPs. Moreover, this pilot study is limited to a single size of a pristine spherical MNP of one plastic polymer. While PS is one of the most highly produced of the major plastic polymers, there are many other highly produced polymers with a wide variety of chemistries, and all MNPs cannot be expected to behave alike in biological systems. Similar studies are therefore needed to investigate intestinal uptake, distribution and maternal–fetal transfer of MNPs across the full range of plastic polymers, which include aliphatics (e.g., polyethylene, polypropylene), haloalkanes (e.g., polyvinylchloride), acrylics (e.g., polymethyl methacrylate) and amides (e.g., Nylon 6).

## 5. Conclusions

This is the first study confirming maternal translocation of ingested MNPs to fetal tissues in a mammalian species. More studies are needed, using environmentally relevant MNPs across a range of highly produced polymers, as discussed above, in order to understand the full scope of potential hazards that ingested MNPs might represent. Further studies are also needed to assess the impact of the presence of MNPs in placental and fetal tissues and cells on fetal health and pregnancy outcomes, in order to uncover the mechanisms involved in MNP uptake by the intestine and in MNP translocation across the maternal–fetal barrier of the placenta, and to determine the role of MNP properties such as polymer type, surface chemistry (e.g., weathering), size and shape on uptake, fetal translocation and health implications.

## Figures and Tables

**Figure 1 nanomaterials-13-00720-f001:**
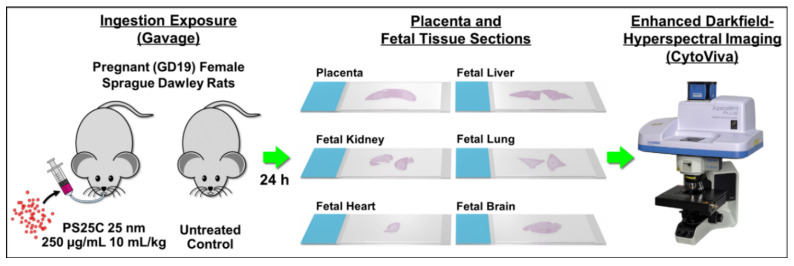
Study design overview.

**Figure 2 nanomaterials-13-00720-f002:**
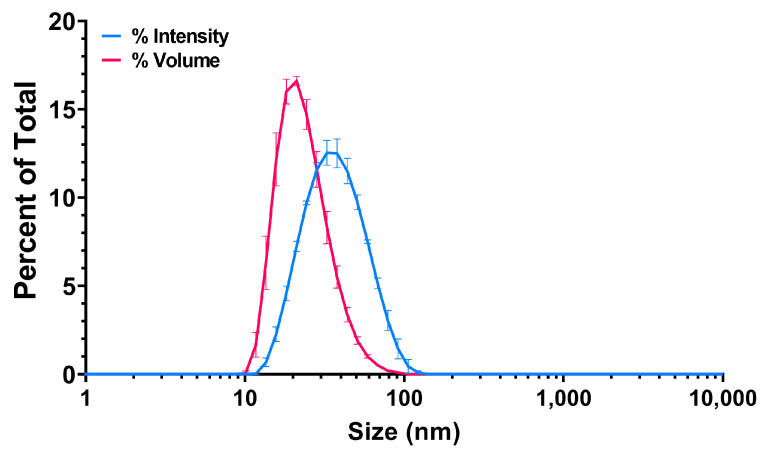
Intensity- and volume-weighted size distributions of carboxylated 25 nm polystyrene (PS25C) dispersed in water.

**Figure 3 nanomaterials-13-00720-f003:**
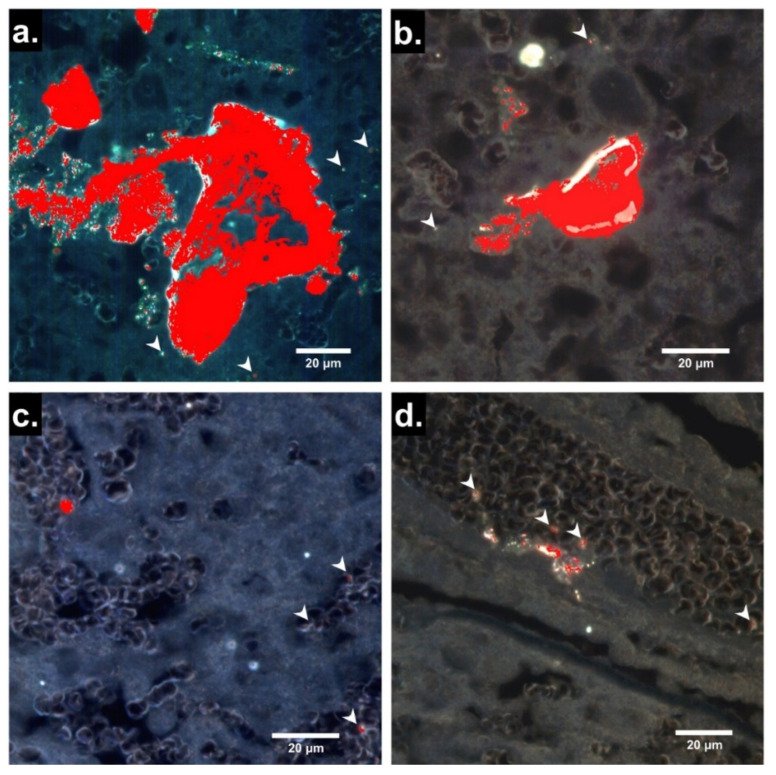
Localization by EDHM of 25 nm carboxylate-modified polystyrene MNPs (PS25C) in rat placenta 24 h after oral gavage. (**a**–**d**) Enhanced darkfield hyperspectral microscopy (EDHM) images of 4 µm sections of placentas from pregnant rats gavaged with 10 mL/kg of 250 µg/mL PS25C in water on GD19 and sacrificed on GD20. Pixels that carry the hyperspectral information of PS25C are overlaid in red. Arrowheads indicate smaller discrete PS25C signals.

**Figure 4 nanomaterials-13-00720-f004:**
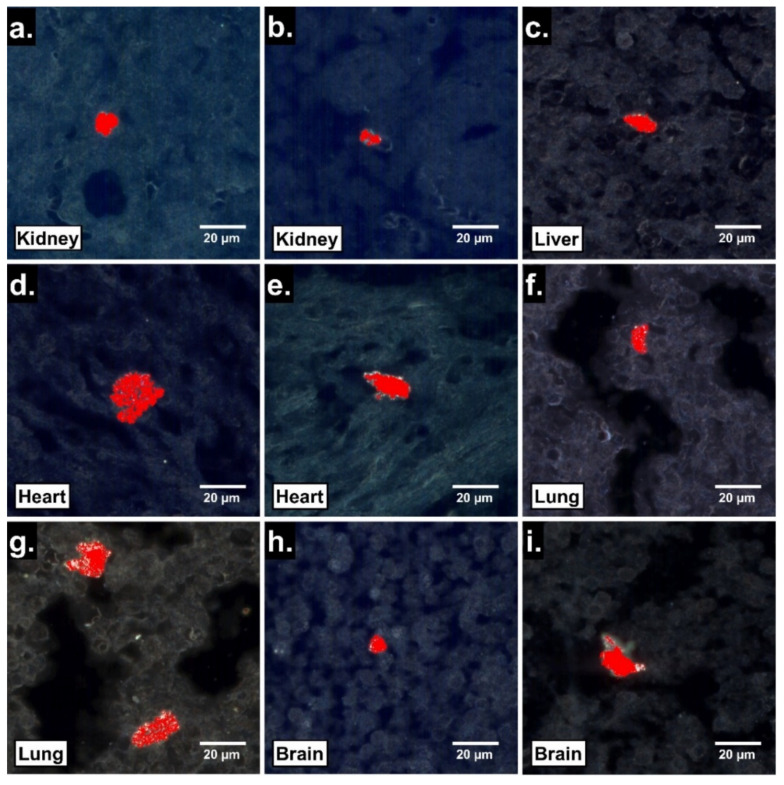
Localization by EDHM of 25 nm carboxylate-modified polystyrene MNPs (PS25C) in rat fetal tissues 24 h after oral gavage in pregnant rats. Enhanced darkfield hyperspectral microscopy (EDHM) images of 4 µm sections of fetal kidney (**a**,**b**), liver (**c**), heart (**d**,**e**), lung (**f**,**g**) and brain (**h**,**i**) from pups of pregnant rats gavaged with 10 mL/kg of 250 µg/mL PS25C in water on GD19 and sacrificed on GD20. Red areas indicate location of hyperspectral signal for PS25C.

**Table 1 nanomaterials-13-00720-t001:** Effect of maternal ingestion exposure to 25 nm carboxylate-modified polystyrene (PS25C) on maternal, placental and litter characteristics.

Treatment	n	Maternal Weight (g)	Number of Fetuses per Litter	Fetal Weight (g)	Placental Weight (g)	Placental Efficiency	Number of Resorption Sites
Control	5	293 ± 53	9.20 ± 1.10	4.61 ± 0.07	0.57 ± 0.01	8.12 ± 0.68	0.60 ± 0.89
PS25C	5	298 ± 24	9.60 ± 1.64	4.62 ± 0.04	* 0.47 ± 0.01	* 9.87 ± 0.66	1.20 ± 1.64

Maternal weight, number of fetuses, placental efficiency and number of resorption sites are presented as mean ± SEM. Values for fetal and placental weights are shown as mean ± SEM. n = number of dams. Statistics were analyzed with Student’s *t*-test (* *p* ≤ 0.05).

## Data Availability

All data from this study will be made available upon written request.

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
