# Peer review of "Ingested Polystyrene Nanospheres Translocate to Placenta and Fetal Tissues in Pregnant Rats: Potential Health Implications"

_nanomaterials, 2023, doi:10.3390/nano13040720_

Round 1

Reviewer 1 Report

Dear editor or authors: MNPs have become a nearly ubiquitous contaminant of both the environment and the food web. Results from previous in vitro and in vivo studies suggest that ingested MNPs can cross the intestinal barrier to access the circulatory system and thence the peripheral tissues. While whether the ingested nanoscale PS MNPs could translocate across the intestinal epithelium to reach the circulation, and subsequently cross the maternal fetal barrier of the placenta to reach to fetal circulation and tissues are still unclear. This study was the first study to clearly confirmed that the maternal translocation of ingested MNPs to fetal tissues in a mammalian species.

1: Just as the author said that this study limited to a single size of MNP. Since there are many other highly produced polymers with a wide variety of chemistries. This part need to add some others study of give some reason or prediction.

2: The result part can be improved by add some tight junction protein to see which of them might charged for the translocation of the intestinal epithelium.

3: Add several reason why the author select MNP instead of other highly produced polymers.

All in all, it is an very important study focus not only on our life, food and environment . We are agree to accept this manuscript after add several introduction and discussion part.

Author Response

1: Just as the author said that this study limited to a single size of MNP. Since there are many other highly produced polymers with a wide variety of chemistries. This part need to add some others study of give some reason or prediction.

Response: We agree that studies of MNPs other than polystyrene are urgently needed. Unfortunately such studies to date are rare or non-existent, and we are not aware of any in vivo studies comparing biodistribution of different polymers, much less fetal translocation. This is the first study that we are aware of revealing translocation of any ingested MNP to the fetus. The reason for this is that polystyrene beads are the only MNPs readily available commercially (with multiple fluorescent markers and in multiple sizes and surface chemistries). They have been used in many cell function and tracking applications for decades, but similar particles of other highly produced polymers have not been available. What is preventing such studies across MNP polymers is a lack of standardized, well-characterized, and environmentally relevant nanoscale test materials suitable for biological studies. Major efforts by our lab an others are underway to develop and obtain further funding to allow more extensive development of environmentally relevant test/reference MNPs of all highly polymers across a range of sizes for biological testing. The reason for predicting that MNPs with different chemistries would have different biological effects is that the surface chemistry and other physicochemical properties of nanoscale particles determine their biointeractions, as noted in the text at the end of the discussion section:

“It is well known that the physicochemical properties of nanoscale materials define their cellular uptake and biodistribution [35, 36].”

2: The result part can be improved by add some tight junction protein to see which of them might change for the translocation of the intestinal epithelium.

Response: We agree that western blot or confocal imaging of ZO-1 in intestinal tissues, or other tight junction proteins, or expression analysis of the same could be useful for understanding the mechanisms of uptake by the intestinal epithelium. Although changes in expression of junction proteins during a single gavage are unlikely (given that it takes on average one hour to translate and transcribe a protein – which is about the length of time the intestine would be exposed in a single gavage), this could play an important role in uptake following prolonged or repeated exposures. We are currently conducting and seeking funding to expand such mechanistic studies, including chemical inhibitor and siRNA knockout in vitro studies and conditional knockouts in vivo, in a systematic fashion across MNP chemistries and properties, to determine the roles of passive transcellular and paracellular diffusion and the various active endocytosis mechanisms in the intestine as a function of MNP chemistry, size, and other properties. Our initial results point to a combination of multiple passive and active mechanisms being involved in MNP translocation, and will be reported in a future paper. We have expanded the discussion of mechanisms in the discussion as follows to clarify these points:

Further studies are needed to determine the specific mechanisms involved in MNP translocation across the intestinal epithelium, which could include passive transcellular and/or paracellular diffusion as well as one or more active (energy-dependent) endocytosis mechanisms, recently reviewed by Rennick et al. [32]. Such studies are underway in our lab and will be reported in a future companion manuscript. Moreover, pregnancy hormones (i.e. progesterone), which would be elevated in the pregnant rat model used in this study, are known to cause smooth muscle relaxation, slowing digestion and prolonging gastrointestinal transit, thereby increasing the likelihood of MNP uptake. Further studies are therefore also needed to assess the role of progesterone and other pregnancy hormones on MNP uptake by the intestine.

3: Add several reason why the author select MNP instead of other highly produced polymers.

Response: We are assuming the reviewer is asking why we selected carboxylated PS MNPs over other MNPs of highly produced polymers. The reason for using carboxylated PS is as mentioned above: it is the only polymer readily available commercially as ready to use micro-nanospheres in various sizes. We agree with reviewer that other  more environmentally relevant MNPs  which are the byproduct of environmental degradation of highly produced polymers are needed. We have revised a section of the discussion to clarify this:

Finally, it is worth noting that simplistic models of MNPs such as the PS used in this study are not environmentally relevant. More environmentally relevant MNPs which are by products of thermal, mechanical and photo degradation of polymers in the environment are needed for future toxicological assessment studies. It is well known that the physicochemical properties of nanoscale materials define their cellular uptake and biodistribution [36, 37]. Such environmentally relevant MNPs may have completely different biokinetics and bioactivity outcomes. For example, weathering (photo-aging) of plastics in the environment results in formation of surface carboxyl and carbonyl functional groups [4], which could dramatically alter the biointeractions of otherwise hydrophobic MNPs. Moreover, this pilot study is limited to a single size of a pristine spherical MNP of one plastic polymer. While PS is one of the most highly produced of the major plastic polymers, there are many other highly produced polymers with a wide variety of chemistries, and all MNPs cannot be expected to behave alike in biological systems. Similar studies are therefore needed to investigate intestinal uptake, distribution, and maternal-fetal transfer of MNPs across the full range of plastic polymers, which include aliphatics (e.g., polyethylene, polypropylene), haloalkanes (e.g., polyvinylchloride), acrylics (e.g., polymethyl methacrylate), and amides (e.g., Nylon 6).

Reviewer 2 Report

The stated results of the work are very enriching and necessary in the given issue, therefore I recommend publishing them in full without corrections.

Author Response

The reviewer requested no corrections. We thank the reviewer for the kind words.

Reviewer 3 Report

The paper on “Ingested polystyrene nanospheres translocate to placenta and fetal tissues in pregnant rats: Potential Health Implications” is interesting and contribute to the already existing data on translocation of plastic nanoparticles in living organism. The current investigation is clearly at the preliminary stage where the authors have shown that PS NPs do get into different cells. There are a few limitations to the current paper. Some of them are listed below.

1.       What would be interesting is to check on the fate of PS nanoparticles, impact on the host cells/tissues and also on the fetus. Are there any toxicity to any of these tissues?

2.       How about a dosimetry investigation? It is highly possible that the starting concentration of PS NPs will have significant impact as well.

3.       It is also important to understand the mechanism of translocation of Ps particles. In fact, the authors have mentioned a few of these items on page 9, but no data were provided to support their claim.

4.       There are no control experiments reported here. The authors should try to conduct both positive and negative control experiments to support their observation.

Overall, the paper is interesting, but need a major revision.

Author Response

  1. What would be interesting is to check on the fate of PS nanoparticles, impact on the host cells/tissues and also on the fetus. Are there any toxicity to any of these tissues?

Response: We agree that further studies are needed to assess the potential impact of PS nanoparticles on host cells and tissues. Unfortunately this is beyond the scope of this  study, which focuses purely on the maternal-fetal translocation of MNPs. We are currently working on more mechanistic studies using both cellular and animal models to address exactly these questions,  using  not only PS nanoparticles but with MNPs of all highly produced polymers. Our findings from these extensive studies will be part of companion papers in the future. We have added the following text to the conclusion section to address the need for such studies:

Further studies are also needed to assess the impact of the presence of MNPs in placental and fetal tissues and cells on fetal health and pregnancy outcomes, to uncover the mechanisms involved in MNP uptake by the intestine and in MNP translocation across the maternal fetal barrier of the placenta, and to determine the role of MNP properties such as polymer type, surface chemistry (e.g., weathering), size, and shape on uptake, fetal translocation, and health implications.

  1. How about a dosimetry investigation? It is highly possible that the starting concentration of PS NPs will have significant impact as well.

Response: We agree that dose will have an important impact on biodistribution and toxicity and that studies along these lines across a range of doses (as well as a range of MNP chemistries, sizes, shapes, and other properties) are urgently needed, but the point of this study was to establish translocation to the fetus of one model MNP, which we believed would occur and did occur at the dose administered, and which we believe should be made known to the toxicology community and public as soon as possible. However, as outlined in the manuscript the dose used in this study is considered relevant based on the published exposure related data currently available. We have left much on the table for future studies, and hope that the results of this study, once in the published domain, will help provide the justification needed for funding of these other studies by our lab and others.

  1. It is also important to understand the mechanism of translocation of PS particles. In fact, the authors have mentioned a few of these items on page 9, but no data were provided to support their claim.

Response: A detailed study of the mechanisms involved in intestinal and placental translocation is clearly needed, but is beyond the scope of this study, as explained above. We have initiated such laborious mechanistic studies but significant additional time will be required to complete them. They will be reported in a future manuscripts. We have added the following text to the discussion to point out the need for such studies:

Further studies are needed to determine the specific mechanisms involved in MNP translocation across the intestinal epithelium, which could include passive transcellular and/or paracellular diffusion as well as one or more active (energy-dependent) endocytosis mechanisms, recently reviewed by Rennick et al. [32]. Such studies are underway in our lab and will be reported in a future companion manuscript. Moreover, pregnancy hormones (i.e. progesterone), which would be elevated in the pregnant rat model used in this study, are known to cause smooth muscle relaxation, slowing digestion and prolonging gastrointestinal transit, thereby increasing the likelihood of MNP uptake. Further studies are therefore also needed to assess the role of progesterone and other pregnancy hormones on MNP uptake by the intestine.

  1. There are no control experiments reported here. The authors should try to conduct both positive and negative control experiments to support their observation.

Response: We believe that the unexposed pregnant rats and their pups are a reasonable  negative control for this study, since the signatures of the PS MNPs are uniquely identified in hyperspectral images of tissues that contain it and did not appear in the corresponding tissues of non-exposed dams and pups. We do not believe that a positive control is necessary, since the only question was the presence or absence of the PS in placenta and fetal tissues of exposed rats. A positive control would presumably involve directly injecting PS MNPs into tissues or treating animals in some way that ensures translocation of the test MNPs to the target tissues.  Neither of these options are very feasible or necessary given that hyperspectral imaging specifically identifies the PS MNPs.

Round 2

Reviewer 3 Report

Thanks for considering my earlier suggestions.